# Genome-Wide Identification of the Glycine-Rich RNA-Binding Protein Genes and Their Expression Analysis upon *Aspergillus flavus* Infection in Groundnut (*Arachis hypogaea*)

Alin M. Jose [1,2], Avuthu Tejaswi [1], Eswaran Kokiladevi [2], Niharika Sharma [3] and Kalenahalli Yogendra [1,*]

1 International Crops Research Institute for the Semi-Arid Tropics, Hyderabad 502324, India; alinmariajose@gmail.com (A.M.J.); tejusmedicine.96@gmail.com (A.T.)
2 Department of Plant Biotechnology, Tamil Nadu Agricultural University, Coimbatore 641003, India; kokiladevi@tnau.ac.in
3 NSW Department of Primary Industries, Orange Agricultural Institute, Orange, NSW 2800, Australia; niharika.sharma@dpi.nsw.gov.au
* Correspondence: yogendra.kalenahalli@icrisat.org

**Abstract:** Aflatoxins, carcinogenic secondary metabolites produced by the *Aspergillus* fungi, pose a significant threat to groundnut, making them susceptible to infection and compromising their quality. Despite extensive breeding programs, the need for more durable resistance in groundnut germplasm remains a major challenge. Targeting susceptible genes favoring *Aspergillus* infection in groundnut could offer a promising strategy for achieving durable resistance. The glycine-rich RNA-binding protein (GR-RBP)-coding genes, known for their involvement in plant hypersensitivity and susceptibility to *A. flavus*, have been studied in model plants. However, there needs to be more understanding of the *GR-RBP* gene family in groundnut. In this study, twenty-three *Arachis hypogaea GR-RBP* (*Ah.GR-RBP*) genes were identified, and the chromosomal location, sub-cellular localization, and regulatory elements in the putative promoter region were analyzed. Expression analysis revealed that *Ah.GR-RBP.1*, *Ah.GR-RBP.12*, *Ah.GR-RBP.3*, and *Ah.GR-RBP.15* showed higher expression in the susceptible genotype. This paper would help to provide knowledge on potential candidate target genes for precise breeding interventions for aflatoxin mitigation in groundnut.

**Keywords:** aflatoxin; secondary metabolites; *Aspergillus flavus*; groundnut; glycine-rich RNA-binding protein

## 1. Introduction

Mycotoxins are toxic compounds produced by certain fungi, primarily in grains and nut crops. Aflatoxin, a naturally occurring mycotoxin produced by *Aspergillus flavus* and *Aspergillus parasiticus*, is hazardous and carcinogenic. Groundnut (*Arachis hypogaea* L.) is one of the crops susceptible to *Aspergillus* infection and aflatoxin contamination. Aflatoxin contamination can directly impact the market competitiveness of groundnut products, global trade, and food safety, resulting in significant financial losses. Despite decades of research on groundnut, achieving success in developing resistant varieties has been limited, partly due to environmental variability and the absence of highly resistant donors in the cultivated species [1]. Furthermore, we need to enhance our understanding of the molecular mechanisms underlying aflatoxin resistance and effectively apply this knowledge in groundnut breeding.

Various mechanisms of groundnut *Aspergillus* resistance have been documented, encompassing resistance to aflatoxin production (AP), pre-harvest aflatoxin contamination (PAC), and in vitro seed colonization (IVSC). These mechanisms have been identified in diverse genetic backgrounds [2]. However, the coexistence of all three resistance mechanisms within a single groundnut genetic background remains to be established. Biotechnological

approaches are crucial in creating genotypes with desired traits in such scenarios. Understanding how plants respond to different pathogens is imperative for identifying the target gene. As *Aspergillus* is a necrotrophic fungus, the host plant likely defends against it by activating various quantitative resistance genes. Additionally, research indicates hypersensitive response (HR) reactions facilitate necrotrophic fungal colonization and increase host plant susceptibility [3]. The recent characterization of gene-for-gene susceptibility, involving the recognition of fungal toxins and conferring plant susceptibility during necrotrophic fungal colonization, has been limited to a few structurally resistance-like (R-like) genes [4]. Within this context, maize expression analysis in response to *A. flavus* infection identified genes encoding glycine-rich RNA-binding proteins (GR-RBPs), with *GR-RBP2* being highly expressed in susceptible maize genotypes compared to resistant ones [5]. Targeting the transcriptional inactivation of these candidate susceptibility-associated genes holds the potential for developing durable resistance through focused breeding interventions.

Glycine-rich proteins (GRPs)/glycine-rich RNA-binding proteins (GR-RBPs) form a superfamily characterized by (Gly)n-X repeats organized in a glycine-rich domain, comprising a conserved glycine-rich domain and an RNA-recognition motif (RRM). Numerous GR-RBPs have been identified across different plant species in the past two decades, with *Arabidopsis thaliana* having eight, *Oryza sativa*—six, *Zea mays*—twenty-three, *Vitis vinifera*—nineteen, *Ipomoea trifida*—nine, *Gossypium arboretum*—thirty-seven, *Gossypium raimondii*—thirty-two, and *Brassica oleracea*—nineteen [6–11]. Expression of *GR-RBP*s in various plants is regulated by both biotic and abiotic stresses when their function is investigated [12]. In Arabidopsis, *AtGRP2* contributes to cold tolerance as an RNA chaperone, and *AtGRP7* plays a role in alleviating water deficit stress [13,14]. Similarly, the expression of *OsGRP1* and *OsGRP4* increased in rice in response to freezing stress, with *OsGRP4* also exhibiting strong sensitivity to heat [15]. However, the involvement of *GR-RBP*s in plant responses to biotic stress is less reported. In maize (*Zea mays*), *GR-RBP2* is prominently expressed in susceptible lines exposed to aflatoxin contamination [5]. Likewise, silencing *CaGRP1* in pepper (*Capsicum annum*) conferred resistance to *Xanthomonas campestris* pv *vesicatoria* (*Xcv*) infection by interacting with CaPIK1 (Receptor-Like Cytoplasmic protein kinase1), suppressing negative regulators of defense responses [16]. Additionally, in Arabidopsis, *AtGRP7* serves as a positive regulator of immunity against *Pseudomonas syringae*, *Pectobacterium carotovorum*, and tobacco mosaic virus while acting as a negative regulator against *Botrytis cinerea* [17]. Despite reports of the involvement of GR-RBPs in various pathogen infections in maize [5] and pepper [16], comprehensive genome-wide phylogenetic and functional studies of GR-RBPs in groundnut are currently lacking.

The genome-wide identification and characterization of candidate genes hold immense significance in advancing our understanding of the genetic basis of various biological processes, diseases, and traits. This comprehensive approach allows for the identification of potential targets for crop improvement. In silico approaches aid in exploring genomic sequences, identifying regulatory elements, predicting gene functions, and assessing structural variations. These methods significantly expedite the identification of target genes for further experimental validation, saving time and resources.

To enhance our understanding of the evolutionary dynamics of GR-RBPs in groundnut and investigate their potential roles in *A. flavus* infection, we identified 23 GR-RBP genes in groundnut (*Ah.GR-RBP*s). Through in silico analyses, we documented their chromosome locations, motif compositions, cis-acting elements, evolutionary relationships, genomic collinearity, and selection pressure. Notably, *GR-RBP2* has previously been implicated in maize Aflatoxin contamination [5]. In this study, we further examined the expression analysis of nine orthologs of *GR-RBP2* during *A. flavus* infection in groundnut. These findings offer valuable insights into the functions of GR-RBP family members and lay the groundwork for utilizing them in developing groundnut lines resistant to aflatoxin contamination.

## 2. Materials and Methods

### 2.1. Identification of GR-RBP Genes in Groundnut

The sequences of 8 GR-RBPs were obtained from the Arabidopsis Information Resource (TAIR) (https://www.arabidopsis.org/index.jsp, accessed on 5 October 2023). These sequences were individually subjected to a blast search in the Legume Information System (LIS) sequence server (https://sequenceserver.legumeinfo.org, accessed on 5 October 2023) to identify corresponding GR-RBPs in groundnut. After removing redundant hits, the sequences of all identified candidate genes were retrieved from Arachismine (https://mines.legumeinfo.org/arachismine/begin.do, accessed on 5 October 2023) within the Legume Information System.

For the prediction of molecular weight (MW), isoelectric point (PI), protein instability index, and grand average of hydrophobicity (GRAVY) of the GR-RBPs, the ExPASy proteomics server (https://www.expasy.org, accessed on 10 October 2023) was utilized [18]. Additionally, the subcellular localization of the GR-RBPs was predicted using DeepLoc 2.0 (https://services.healthtech.dtu.dk/services/DeepLoc-2.0, accessed on 10 October 2023) [19].

### 2.2. Phylogenetic Analysis

Groundnut GR-RBPs were classified based on comparisons with GR-RBPs identified in Arabidopsis, maize, Medicago, soybean, and pigeonpea. Multiple sequence alignments for all GR-RBPs were conducted using ClustalW tool in MEGA v11 software. The phylogenetic tree was constructed utilizing the neighbor-joining method (NJ) with 1000 bootstrap replications through MEGA v11 software [20]. Further modifications to the tree were implemented using iTOL v6 (https://itol.embl.de, accessed on 19 October 2023) [21].

### 2.3. Chromosomal Location, Synteny, and Evolutionary Analysis of GR-RBP Genes in Groundnut

Information regarding chromosomal locations was sourced from ArachisMine (https://mines.legumeinfo.org/arachismine/begin.do, accessed on 5 October 2023) in LIS and was employed to map the identified genes onto their respective chromosomes using the online tool MapGene2Chromosome v2.1 (http://mg2c.iask.in/mg2c_v2.1/, accessed on 19 October 2023) [22].

All groundnut data utilized in synteny analysis were retrieved from ArachisMine (https://mines.legumeinfo.org/arachismine/begin.do, accessed on 5 October 2023) in LIS, while data for Arabidopsis, maize, Medicago, soybean, and pigeonpea were obtained from TAIR, Phytozome, and LIS, respectively. Synteny analysis was conducted using the Multiple Collinearity Scan toolkit (McScanX) in TBtools v2.030 with CPU parameters for BlastP-2, E-value: 1e-10, and the number of BlastHits-5 [23]. The genomes and annotation files of the respective crops under analysis were prepared and utilized as input for analysis through the One Step MCScanX Wrapper function. Subsequently, the synteny plot was generated using the Multiple Synteny Plot function in TBtools.

### 2.4. Analysis of Intron-Exon Structure and Conserved Domains

The coding and genomic sequences of the identified candidate *GR-RBP* genes in groundnut were employed to discern the gene structure, utilizing Gene Structure Display Server 2.0 (http://gsds.gao-lab.org, accessed on 26 October 2023) [24]. To identify conserved domains, HMMER (https://www.ebi.ac.uk/Tools/hmmer, accessed on 26 October 2023) was employed [25].

### 2.5. Cis-Regulatory Element Analysis of Ah.GR-RBPs

The term "sequences of the proximal promoter region" pertains to sequences located 2000 base pairs upstream of the transcriptional start sites. Groundnut promoter sequences for *GR-RBP* genes were obtained from the Arachismine database within LIS. For the analysis of cis-regulatory elements, all promoter sequences were submitted to the Plant-CARE database (http://bioinformatics.psb.ugent.be/webtools/plantcare/html, accessed on

26 October 2023) [26]. The schematic diagram illustrating the cis-regulatory elements was generated using TBtools.

*2.6. Protein-Protein Interaction*

Examining protein–protein interactions is crucial for understanding the molecular regulatory system. The protein–protein interactions of Ah.GR-RBPs were anticipated using the STRING online tools (http://string-db.org, accessed on 26 October 2023) [27]. Orthologs of the submitted proteins were determined in *Zea mays* through BLAST, and the ortholog with the highest bit score was subsequently explored. The non-interacting predicted proteins were removed.

*2.7. Expression Analysis of GR-RBPs during A. flavus Infection*
2.7.1. Plant Material

Groundnut seeds representing the resistant genotype (55-437) and the susceptible genotype (TMV-2) were procured from the groundnut breeding unit at ICRISAT, India. The 55-437 genotype, a drought-tolerant Spanish type, was developed through pedigree selection from plant materials received from Hungary at the National Centre for Agronomic Research in Bambey, Senegal. Meanwhile, the TMV-2 Spanish-type groundnut, a susceptible genotype, was derived by mass selection from the Gudiatham bunch (AH-149 32) at the Department of Oilseeds, Tamil Nadu Agricultural University, Coimbatore, India. These specific genotypes were chosen based on their consistent low and high aflatoxin contamination observed in previous studies [28].

The stored mature seeds from both genotypes underwent surface sterilization using 0.1% mercuric chloride, were washed, and soaked for 4 h. Subsequently, the de-embryonated cotyledons were halved and placed in Petri dishes containing sterile agar. *A. flavus* toxigenic strain AF 11-4, obtained from the groundnut pathology collections at ICRISAT, India, was used for infection [5,29]. The fungal cultures were grown on Potato Dextrose Broth (PDB) medium at 30 °C in the dark and maintained as 30% glycerol stocks at −80 °C. For inoculum preparation, the fungus was multiplied on soaked and autoclaved peanut seeds, to which 5 mL of *A. flavus* spore suspension was added. These were incubated at 30 °C for 4–5 days to allow sporulation. Virulent spores were collected in sterile distilled water containing 0.05% Tween 20 and diluted to a concentration of $5 \times 10^4$ spores/mL using a Neubauer hemocytometer and the colony-forming units (CFUs) determined by standard 10-fold dilutions to obtain ~40,000 cfu/mL on *Aspergillus flavus* and *parasiticus* agar (AFPA) medium. The prepared spore suspension was inoculated on each of the diced portions of the cotyledons, and the samples were incubated for 48 h [30]. The resulting samples were labeled as follows: resistant mock (RM) for the ICGV 55-437 genotype with water inoculation, resistant pathogen (RP) for the ICGV 55-437 genotype inoculated with *A. flavus* spores, susceptible mock (SM) for the TMV2 genotype with water inoculation, and susceptible pathogen (SP) for the TMV2 genotype inoculated with *A. flavus* spores. Following inoculation, the cotyledons were incubated at 37 °C for 48 h, and were collected and frozen in liquid nitrogen and stored at −80 °C until utilized for gene expression studies.

2.7.2. RNA Isolation and Expression Analysis

Following the manufacturer's instructions, RNA was isolated from infected and non-infected groundnut cotyledons using the RNeasy Plant Mini kit (Qiagen, Tokyo, Japan). The integrity of the extracted RNA was then evaluated through 2% agarose gel electrophoresis, and the quality of the RNA samples was determined using an Invitrogen Qubit fluorometer (Fisher Scientific, Loughborough, UK).

For the subsequent quantitative reverse-transcription polymerase chain reaction (qRT-PCR), the first-strand cDNA synthesis was carried out utilizing 2.0 μg of pure RNA, following the specified methodology (Invitrogen, Carlsbad, CA, USA; Thermoscript RT-PCR equipment). The quality and integrity of the resulting cDNA were verified through 1% agarose gel electrophoresis. Among the 23 *GR-RBP* genes identified in groundnut,

9 genes that belong to the GR-RBP2 family were selected, gene-specific primers (Table S1) were designed, and quantitative PCR was performed with a CFX96TM Real-Time System (Bio-Rad, Gurugram, India). The groundnut *ADH3* (alcohol dehydrogenase) gene served as a housekeeping gene for normalizing cycle threshold (Ct) data for all three biological replicate samples with each of the two technical replications [31].

## 3. Results

### 3.1. Identification of GR-RBP Genes in Groundnut

In the quest to identify coding genes for glycine-rich RNA-binding proteins (GR-RBPs) in the groundnut genome, all eight GR-RBP family genes from *Arabidopsis thaliana* were utilized as queries in a BLAST search against the *Arachis hypogaea* genome archived in the Legume Information System (LIS) (https://sequenceserver.legumeinfo.org, accessed on 5 October 2023). Following the comparison, redundant genes were eliminated, revealing the presence of 23 distinct copies of the GR-RBP gene family across the groundnut genome. Despite their shared membership in the GR-RBP family, these proteins exhibited notable variations in predicted physicochemical properties. The open reading frame (ORF) spanned from 222 to 1218 bp, and the amino acid length ranged from 73 to 405 aa (amino acids). The protein specifications also displayed diversity, with molecular weights ranging from 8 kDa to 41 kDa and isoelectric points (PI) spanning from acidic (4.42) to basic PI (11.53). Most of the proteins were predominantly localized in the mitochondrion (13) and cytoplasm/nucleus (7), while a smaller subset was found in the plastid (2) and endoplasmic reticulum (1) (Table S2).

### 3.2. Phylogenetic Analysis of GR-RBPs in Groundnut

To explore the phylogenetic relationships within the GR-RBP family, reference sequences from model plants and various closely related species, such as Arabidopsis, maize, Medicago, soybean, and pigeonpea, were examined. The *GR-RBP* genes were categorized into three clades based on the similarity index among their protein sequences (Figure 1). The largest clade, Clade III, encompassed 32 homologous sequences from all the species clustered together with a bootstrap value of 100%. Following this, Clade II comprised 25 homologous sequences clustered with a bootstrap value of 29%. Clade I included only two sequences of *GR-RBP* genes clustered together with a bootstrap value of 100%. The clustering of all GR-RBP family genes from different species into three clades suggests that the gene family is primarily conserved among these species and has undergone considerably fewer evolutionary changes.

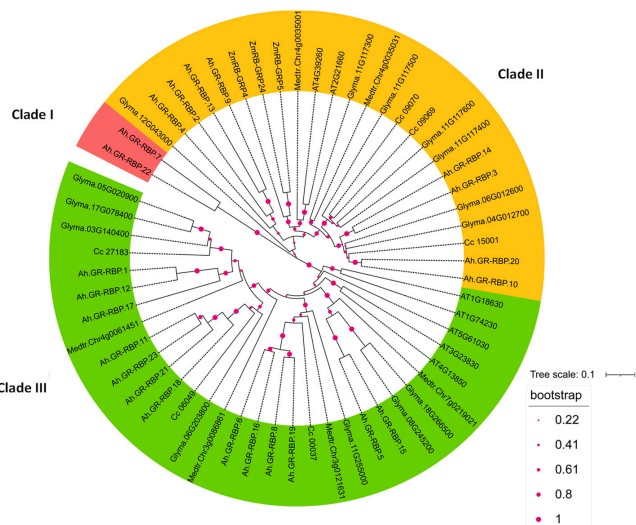

**Figure 1.** The phylogenetic tree of *GR-RBPs* genes uses MEGA11 by the neighbor-joining method with 1000 bootstrap replications. The tree was classified into 3 different clades indicated by different colors.

### 3.3. Chromosomal Location, Synteny, and Evolutionary Analysis of GR-RBP Genes in Groundnut

The analysis of chromosomal localization for *GR-RBP* genes in the groundnut genome revealed a distribution where three genes are located on chromosomes 4 and 14. In comparison, two genes are present on chromosomes 1, 6, 13, and 19. Additionally, one gene each is localized on chromosomes 3, 5, 7, 10, 11, 15, 16, 17, and 20 (Figure 2).

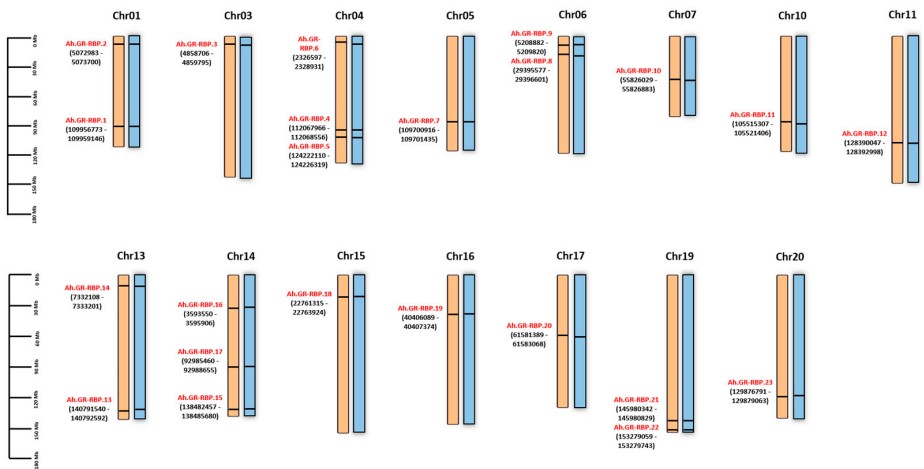

**Figure 2.** Chromosomal locations of GR-RBP genes in groundnut depicts the locations of GR-RBP genes on sub-genome A and B. The Scale on the left side represents the length of the chromosome in Mb. Orange color indicates A sub-genome and blue color indicates B sub-genome.

Synteny analysis, aimed at understanding the evolutionary relationship of *GR-RBP* genes in groundnut concerning Arabidopsis and maize, uncovered a collinear relationship among these species (Figure 3). In Arabidopsis, six homologous *GR-RBP* genes were duplicated into eight copies that exhibited collinearity with groundnut. In contrast, one *GR-RBP* gene in maize was duplicated into three collinear copies with groundnut. To delve deeper into the evolutionary dynamics of *GR-RBP* genes, calculations of Ka (non-synonymous substitutions per site), Ks (synonymous substitutions per site), and Ka/Ks ratios were conducted for each duplication pair (Table S3). Ka/Ks ratios less than one indicate that these genes have undergone purifying selection following gene duplication events.

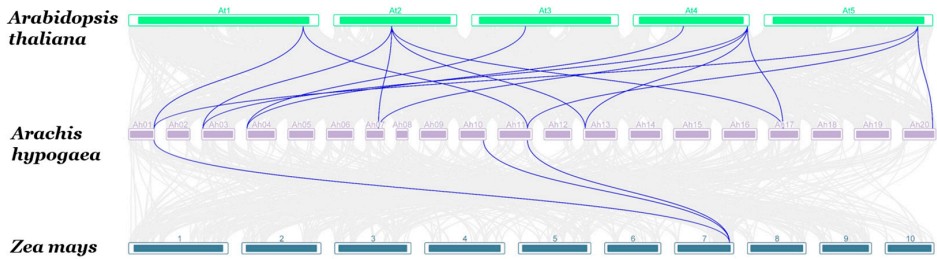

**Figure 3.** Representation of collinearity and synteny analysis of *GR-RBP* genes among *Arabidopsis thaliana*, *Arachis hypogaea*, and *Zea mays*. The *GR-RBP* genes in *Arachis hypogaea* collinear with other crops are represented in blue lines. The chromosome numbers of each crop are represented by the numbers above each chromosome respectively.

### 3.4. Analysis of Intron–Exon Structure and Conserved Domains

Examining gene structure to identify structural diversity revealed variations in the number of exons for each gene, ranging from one to six (Figure S1). Through conserved domain prediction, it was observed that RNA recognition motifs (RRMs) were present in all except *Ah.GR-RBP.7* and *Ah.GR-RBP.22*. Most RRMs displayed high similarity with two consensus RNP regions and glycine-rich residues (Figures 4 and S2).

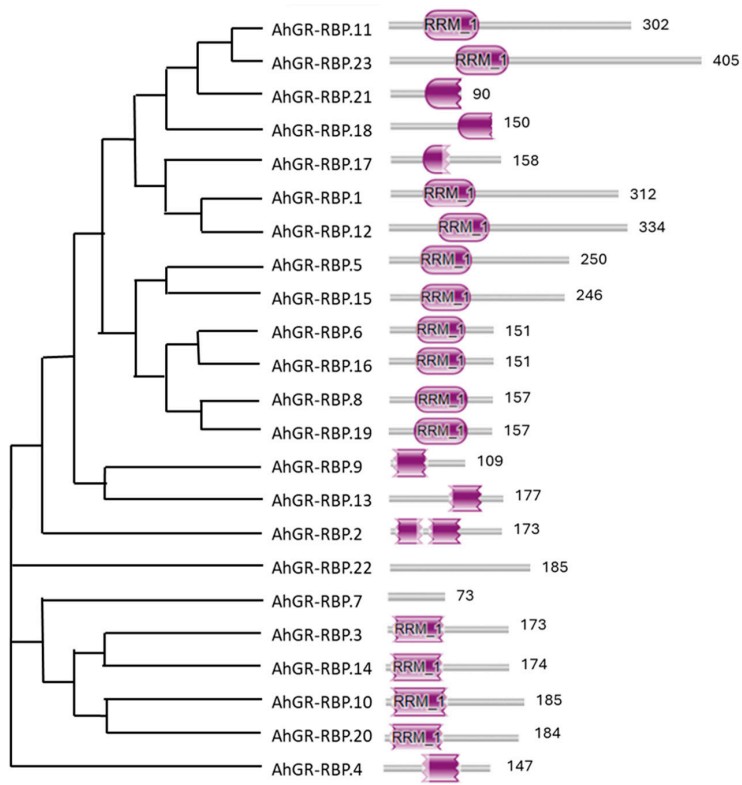

**Figure 4.** Domain analysis of GR-RBPs from groundnut. The purple-colored boxes represent the RRM (RNA recognition motif) domain. The GR-RBPs are given in the order of the phylogenetic tree.

### 3.5. Cis-Regulatory Element Analysis of Ah.GR-RBPs

Previous investigations have underscored the crucial role of promoter cis-acting elements in governing gene expression. Utilizing the PlantCARE online database, this study delved into the analysis of cis-elements, revealing the presence of various factors associated with abiotic stress, such as heat, cold, salt, drought, and ABA, which are also implicated in hormones and stress responses. In addition to fundamental cis-elements like CAAT-box and TATA-box, a diverse array of regulatory elements, encompassing hormone- and light-responsive elements, development- and stress-related elements, site-binding-related elements, and others, identified in all promoters.

Within this spectrum of elements, particular attention was given to those explicitly linked to stress to ascertain their role in aflatoxin contamination in groundnuts. Stress-responsive elements, such as dehydration-responsive elements (DRE core), low-temperature-responsive elements (LTR), anaerobic-responsive elements (ARE), and those responsive to biotic stresses (W-box, TC-rich repeats, G Box, and STRE), along with drought-inducible MYB binding sites (MBS), MYB, MYC, ACTCATCCT, AP1, WUN-motif, etc., were identified in the presumed gene promoter regions of *GR-RBP* genes in groundnut genomes. In total, 20 stress-responsive elements have been identified in the groundnut promoter region of *GR-RBPs*. The quantity of these elements in each gene promoter is detailed (Figure 5). Notably, biotic stress-responsive elements like STRE, G-Box, TC-rich repeats, and W-box were discerned in the promoter regions of the *GR-RBP* genes in groundnut. Identifying stress-related elements, especially the biotic stress-related elements in the promoter region of the *GR-RBP* genes, implies a crucial role for these genes in conferring biotic stress tolerance in groundnut.

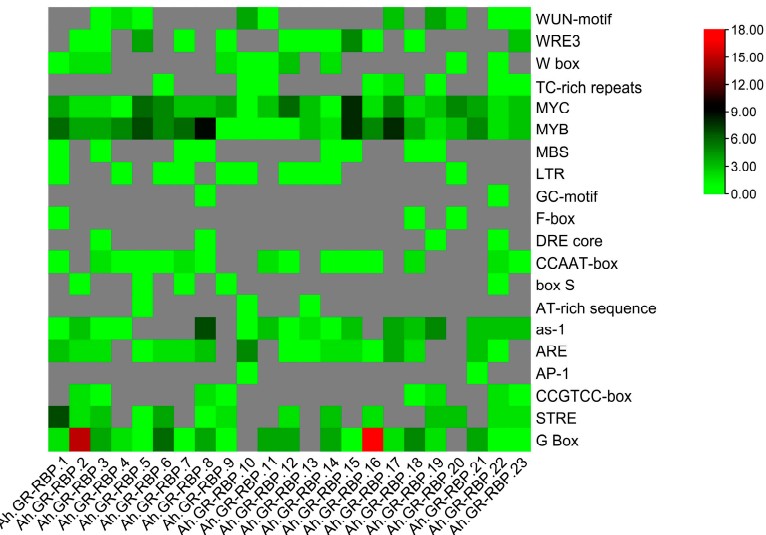

**Figure 5.** The distribution of stress-related cis-elements in the groundnut promoter region of GR-RBP genes.

*3.6. Protein–Protein Interaction of GR-RBPs in Groundnut*

The cellular machinery is predominantly structured around proteins and their functional interactions, and a comprehensive understanding of biological events necessitates consideration of their network connections. In this study, the interaction network of Ah.GR-RBPs were employed to elucidate how these proteins may interact with various others. The interactive network (Figure 6) revealed that Ah.GR-RBP.19, Ah.GR-RBP.20, Ah.GR-RBP.22, and Ah.GR-RBP.23 engages in interactions with other proteins. Notably, it was identified that most interacting proteins are associated with RNA metabolism and exhibit responses to diverse stimuli such as stress and hormones. Additionally, one interacting protein, B6STS5 (RGA4), was implicated in plant disease resistance. RGA4 (R gene analogs 4) is a mitochondrial carrier-like protein crucial in ADP binding [32]. RGA4 represents a CN (coiled-coil NBS) type of R gene capable of providing disease resistance.

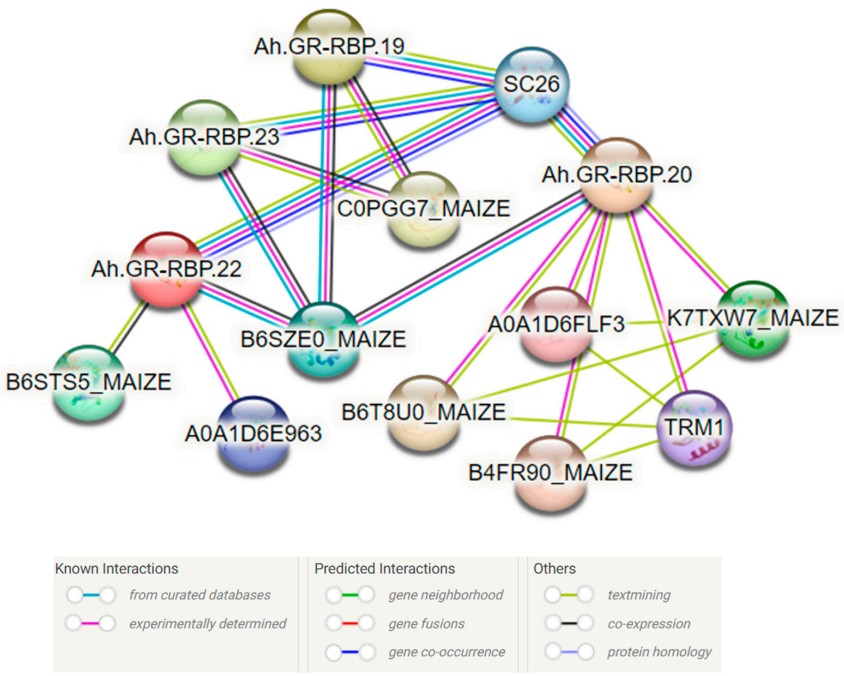

**Figure 6.** Predicted protein interaction networks of GR-RBPs. The green line is neighborhood evidence; the blue line is co-occurrence evidence; the purple line is experimental evidence; the yellow line is text mining evidence; the light blue line is database evidence; the Black line is co-expression evidence.

*3.7. Expression Analysis of GR-RBPs during A. flavus Infection*

To explore the responsiveness of *Ah.GR-RBP*s to biotic stress, particularly during *A. flavus* pathogen infection, nine genes were selected for expression profile analysis from twenty-three genes of the GR-RBP family identified in groundnut. These nine genes, in particular, belong to GR-RBP2, reportedly involved in maize–*A. flavus* interaction. Therefore, the targeted set of gene analysis was conducted under treatment with the *A. flavus* pathogen on both resistant (ICGV 55-437) and susceptible (TMV2) genotypes. The experimental design incorporated three biological replicates and two technical replicates of each susceptible, resistant, mock, and pathogen-inoculated sample.

The expression patterns of the nine *Ah.GR-RBP*s exhibited variation among the samples. Specifically, the genes *Ah.GR-RBP.6* (0.57- and 0.70-fold change), *Ah.GR-RBP.16* (0.50 and 0.57 FC), *Ah.GR-RBP.5* (0.44 and 0.67 FC), *Ah.GR-RBP.8* (0.16 and 0.80 FC), and *Ah.GR-RBP.19* (0.18 and 0.35 FC) were downregulated in RP and SP samples compared to their mock samples. In contrast, *Ah.GR-RBP.12* (17 FC), *Ah.GR-RBP.1* (42 FC), *Ah.GR-RBP.3* (3.61 FC), and *Ah.GR-RBP.15* (52 FC) exhibited upregulation in SP compared to RP samples (Figure 7). These findings suggest a potential role for these genes in *A. flavus* infection in groundnut.

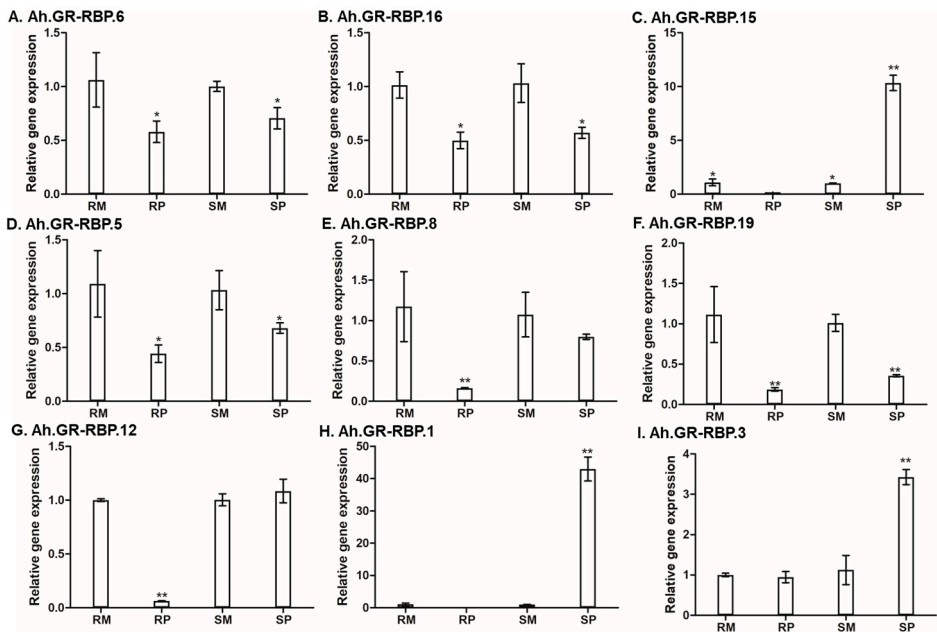

**Figure 7.** Relative transcript expression of contrasting groundnut genotypes during *A. flavus* infection, based on qRT–PCR in comparison to reference gene *ADH*. (**A**). *Ah.GR-RBP.6*, (**B**). *Ah.GR-RBP.16*, (**C**). *Ah.GR-RBP.15*, (**D**). *Ah.GR-RBP.5*, (**E**). *Ah.GR-RBP.8*, (**F**). *Ah.GR-RBP.19*, (**G**). *Ah.GR-RBP.12*, (**H**). *Ah.GR-RBP.1*, (**I**). *Ah.GR-RBP.3*. RP is a resistant genotype with *A. flavus* inoculation, RM is a resistant genotype with mock inoculation, SP is a susceptible genotype with *A. flavus* inoculation, and SM is a susceptible genotype with mock inoculation. * $p < 0.05$, ** $p < 0.01$, a significant difference in expression level in RP compared to SP.

## 4. Discussion

GR-RBPs represent a distinct subgroup of class IV glycine-rich RNA-binding proteins within the glycine-rich (GRPs) superfamily [33]. Extensive research has been conducted on numerous *GR-RBP* genes across various species, including wheat, tobacco, rice, maize, Arabidopsis, apple, sweet potato, and grapes. This is attributed to their pivotal roles in regulating gene expression at post-transcriptional levels [7–9,34–38]. GR-RBPs have been recognized for their involvement in crop plants' biotic and abiotic stresses [39]. Despite the extensive study of *GR-RBP* genes in other plant species, their groundnut characterization still needs improvement. Therefore, our efforts focused on identifying *GR-RBP* genes in

groundnut, exploring their evolutionary relationships, comparing their sequence features, and analyzing their expression patterns in cotyledons infected with *A. flavus*.

### 4.1. Identification and Diversity Analysis of GR-RBPs in Groundnut

We have identified twenty-three *GR-RBP* genes in groundnut through sequence and conserved domain analysis. Plant GR-RBPs are characterized by possessing an RNA-binding domain, typically comprising an RNA recognition motif (RRM) or a cold shock domain (CSD) at the N-terminal. At the same time, the C-terminal exhibits a high glycine content ranging from 20% to 70%. In particular, glycine constitutes 67% of the amino acids in the GRP-1 protein [40]. The multiple sequence alignment results further revealed the presence of glycine-rich regions and two highly conserved RNP (ribonucleoprotein) motif sequences, namely RNP1 and RNP2. The RRM domains encompass RNP1, eight amino acid residues, and RNP2, six amino acid residues. In our study, we observed that some of the Ah.GR-RBPs among the twenty-three identified exhibit a lack of conservation in the RRM motif and glycine-rich residues. This analogous trend was also noticed in previous studies on GR-RBPs identified in maize and grapes despite their classification within the GR-RBP family of proteins [7,8].

Moreover, the gene structure of *Ah.GR-RBP* genes displays significant variation, with the number of exons ranging from one to six. Notably, *Ah.GR-RBP.11* has been reported to have no introns. Predictions of subcellular localization for Ah.GR-RBP proteins indicated that most were located in the mitochondrion, although seven Ah.GR-RBPs were found in the cytoplasm/nucleus. Additionally, Ah.GR-RBP.7 and Ah.GR-RBP.22 were localized to the plastid, while Ah.GR-RBP.13 was identified in the endoplasmic reticulum. Previous studies reported their presence in mitochondria, cytoplasm, nuclear regions, and even the vacuole for AtGRP5 [33,41,42]. This diverse cellular localization aligns with their role as RNA-binding proteins, suggesting involvement in pre-mRNA processing. Earlier studies demonstrated that several GR-RBPs exhibit RNA-binding activity in vitro, binding single-stranded DNA (ssDNA) and undergoing phosphorylation in vivo and in vitro [43–45]. Deletion experiments on RZ-1 highlighted the essential role of the glycine-rich and RNA-binding domains in RNA binding [46]. These findings suggest that GRPs may play a role in the maturation, processing, or regulation of gene expression of RNA.

### 4.2. Evolution of GR-RBP Genes in Groundnut

Phylogenetic analysis confirmed the evolutionary connection between *Ah.GR-RBP*s and known GR-RBPs from Arabidopsis, maize, Medicago, soybean, and pigeonpea. Our study identified three clades, aligning with the distribution of GR-RBPs in grapes that also form three clades [8]. In contrast, GR-RBPs in other crops like maize, sweet potato, cotton, and broccoli are often distributed into four clades [7,9–11]. This comparative genomics approach strengthens the evidence for their shared ancestry. The exon structure of these genes further contributes to understanding their evolutionary relationships.

In response to the dynamic environmental challenges, crop plants duplicate their genes, and the duplicated genes often undergo sub- or neo-functionalization [47]. Redundant copies from this process may exhibit similar functionalities or function redundantly, which is essential for functional divergence or conservation [48,49]. The obtained homologous genes between groundnut and Arabidopsis outnumbered those with maize, indicating a closer phylogenetic and evolutionary relationship between groundnut and dicotyledons than with monocotyledons. The collinearity observed among these homologous genes in groundnut suggests a significant expansion of the *GR-RBP* gene family during genome evolution, further complicated by the tetraploid nature of groundnut. Additionally, the Ka/Ks ratios point towards the substantial influence of purifying selection in the evolution of *GR-RBP* genes, emphasizing their likely role in environmental adaptability.

### 4.3. Regulation of GR-RBPs in Groundnut

Cis-regulatory elements play a crucial role in gene expression regulation, participating in various essential cellular processes [50]. The promoter region, located 2000 base pairs upstream of the transcription start site, harbors these elements. Unlike coding sequences, cis-acting elements serve as binding sites for proteins and regulatory factors, such as transcription factors, influencing gene transcription and expression. Therefore, exploring diverse cis-elements is vital for further functional investigations into GR-RBPs.

Our analysis identified twenty predicted cis-regulatory elements associated with stress [51]. Biotic stress-related cis-elements include W-box, G-box, TC-rich repeats, and STRE [52,53]. The G-box, which is considered a pathogen-inducible element, is ubiquitous and crucial for regulating biotic and abiotic stress. TC-rich repeats are involved in defense and stress responses, while STRE (stress-responsive element) is associated with stress response. The W box (TTGACC) is an elicitor-responsive cis-element that interacts with WRKY family transcription factors, functioning in various stressors, seed dormancy, senescence, and defense-related gene (PR gene) expression control. These stress-related cis-elements underscore the significance of *GR-RBP*s in *A. flavus* infection. Besides biotic stress-related elements, a relative abundance of abiotic stress-responsive (MYB, LTR, DRE core, etc.) elements indicate the significant functional role of *GR-RBPs* in several abiotic stress responses and tolerance in groundnut.

Protein–protein interactions (PPIs) form complex networks essential for diverse biological processes. Analyzing PPIs provides insights into the physical connections between Ah.GR-RBPs and other proteins. Among the identified GR-RBPs in groundnut, only four were found to interact with other proteins. Most interacting proteins were associated with RNA metabolism, indicating the role of GR-RBPs as RNA-binding proteins. Additionally, other interacting proteins were linked to stress-related responses. Notably, RGA4, belonging to the CC-NBS family, was involved in disease resistance, suggesting a potential role for GR-RBPs in plant defense during pathogen attacks, including *A. flavus* infection [54].

### 4.4. Expression Analysis of GR-RBP Genes in Groundnut in Response to A. flavus Infection

*GR-RBP*s in Arabidopsis, rice, and maize have been implicated in stress response mechanisms [6,7,55,56]. To elucidate the involvement of *GR-RBP*s in mitigating aflatoxin contamination, we conducted expression analysis on nine *Ah.GR-RBP*s under *A. flavus* infection. Previous research has highlighted the significance of these *GR-RBP*s in conferring tolerance to biotic stress. For instance, *GR-RBP2* was notably upregulated in a susceptible maize genotype in response to *A. flavus* [5]. Similarly, *GR-RBP1* (*CaGRP1*) in pepper exhibited negative regulation in response to the *Xanthomonas campestris* pv *vesicatoria* (*Xcv*) pathogen [16]. To elucidate the involvement of *GR-RBP*s in mitigating aflatoxin contamination in groundnuts, we conducted expression analysis on nine *Ah.GR-RBPs* that belong to GR-RBP2 which were inferred to be upregulated in maize in response to *A. flavus* infection. Our study focused on cotyledons, as *A. flavus* primarily targets groundnut cotyledons. The qRT-PCR expression analysis showed no significant difference in expression for *Ah.GR-RBP.6*, *Ah.GR-RBP.16*, *Ah.GR-RBP.5*, *Ah.GR-RBP.8*, and *Ah.GR-RBP.19* between RP and SP. However, *Ah.GR-RBP.1*, *Ah.GR-RBP.12*, *Ah.GR-RBP.3*, and *Ah.GR-RBP.15* genes were significantly upregulated in SP samples compared to RP samples. These findings align with maize studies, suggesting that *GR-RBP*s act as negative regulators of immunity against *A. flavus* infection [5]. Previous reports also underscore the role of *GR-RBP*s in conferring resistance to pathogens. In Arabidopsis, the GR-RBP-encoding gene, TaRZ1, expression increased resistance against *Pseudomonas syringae* DC3000 pathogenic bacteria [57]. Similarly, *At-GRP7* in Arabidopsis functioned as a positive regulator of plant immunity against *Pseudomonas syringae*, interacting with the Type III effector molecule, and also displayed resistance against *Pectobacterium carotovorum* and tobacco mosaic virus [17,58]. However, *At-GRP7* acted as a negative regulator of immunity against *Botrytis cinerea* [17]. This dual role of *GR-RBP* as both a positive and negative regulator of plant immunity led

us to hypothesize that *GR-RBP* could serve as a potential target for developing groundnut genotypes resistant to *A. flavus*.

## 5. Conclusions

In our investigation, we identified 23 *Ah.GR-RBP*s across the groundnut genome through various bioinformatic analyses. Additionally, we delved into gene structure analysis, chromosome location, conservative domain analysis, and synteny analyses. The expression analysis of the nine *Ah.GR-RBP*s exhibited variations between *A. flavus*-inoculated and uninoculated samples of resistant and susceptible groundnut genotypes. Specifically, *Ah.GR-RBP.1*, *Ah.GR-RBP.12*, *Ah.GR-RBP.3*, and *Ah.GR-RBP.15* displayed higher expression in susceptible genotypes upon *A. flavus* infection. These genes emerge as potential targets for precision breeding in groundnuts to enhance resistance against *Aspergillus* and mitigate aflatoxin contamination. Our findings offer a new perspective on the significance of GR-RBPs in groundnut and their role in *A. flavus* infection.

**Supplementary Materials:** The following supporting information can be downloaded at: https://www.mdpi.com/article/10.3390/agronomy14010165/s1, Figure S1: Exon/intron distribution in the *GR-RBP*s in groundnut. The neighbor-joining tree of Ah.GR-RBPs were constructed using MEGA11 software. The intron and structures (gene structure) of *Ah.GR-RBP*s were displayed using the Gene Structure Display Server. Figure S2: Alignment of GR-RBPs amino acid sequences from groundnut. The conserved amino acids are in green and predicted purple lines mark conserved domains. The RNP-1, RNP-2 domains, and glycine-rich regions are indicated in the sequence. Table S1: List of qPCR primers used in this study. Table S2: Detailed information GR-RBPs of groundnut corresponding gene ID, coding sequences, chromosome location, number of exons, subcellular localization, and predicted physiochemical properties. Table S3: Ka/Ks ratios between GR-RBPs between groundnut and Arabidopsis.

**Author Contributions:** Conceptualization: E.K. and K.Y.; methodology: A.M.J., A.T. and K.Y.; software: A.M.J., A.T. and N.S.; data analysis: A.M.J., A.T. and K.Y.; writing—original draft preparation: A.M.J.; writing—review and editing: K.Y., N.S. and E.K.; supervision and funding acquisition: K.Y. All authors have read and agreed to the published version of the manuscript.

**Funding:** This work was carried out with the aid of a grant from the Start-up Research Grant (SRG) (File No. SRG/2021/000422) from the Science and Engineering Research Board (SERB), Govt. of India.

**Data Availability Statement:** Data will be available from the corresponding author upon reasonable request.

**Acknowledgments:** AMJ is supported by the ICAR Junior/Senior Research Fellowship ICAR-JRF/SRF (PB05076080183), Government of India, for her doctoral research work.

**Conflicts of Interest:** The authors declare no conflicts of interest.

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
