# Peer review of "Genome-Wide Identification of the Glycine-Rich RNA-Binding Protein Genes and Their Expression Analysis upon Aspergillus flavus Infection in Groundnut (Arachis hypogaea)"

_agronomy, doi:10.3390/agronomy14010165_

Round 1
Reviewer 1 Report
Comments and Suggestions for Authors
This is a well written manuscript providing the first report of the expression of GR-RBP genes in peanut infected with A. flavus.
These results should provide a significant advance in breeding for resistance and should be of interest to many other scientists.
Author Response
January 1st, 2024
Dear Editor
Thank you very much for your valuable comments on our manuscript “Genome-wide identification of GR-RBP genes and their expression analysis upon Aspergillus flavus infection in groundnut (Arachis hypogaea)” by Jose et al. We have modified our manuscript as per the reviewers’ comments, and a point-wise response to the queries is appended below.
Sincerely,
Kalenahalli Yogendra
REVIEWER 1:
This is a well-written manuscript providing the first report on the expression of GR-RBP genes in peanuts infected with A. flavus.
These results should provide a significant advance in breeding for resistance and should be of interest to many other scientists.
Response: Thank you for your positive feedback on our manuscript.
Reviewer 2 Report
Comments and Suggestions for Authors
Dear Author,
I have an honor to review the manuscript entitled “Genome-wide identification and expression analysis of GR[1]RBPs during Aspergillus flavus infection in groundnut (Arachis hypogaea)” a research article submitted to the MDPI Journal, Agronomy. Authors of this manuscript documented GR-RBP gene families in groundnut using in silico based genome-wide scanning analysis. They have identified chromosomal location, sub-cellular localization, and types of regulatory elements present in the putative promoter region. Further performed expression analysis of the some of the selected genes after A. flavus infection. Overall, the experiments are performed well and the results are convincing. Thus, the presented results take up an important topic consistent with the profile of the Journal.
-However, even, manuscript is well organized and well described of the conception; I have some suggestions, which might improve the manuscript to make important to the wider audience.
-English should be improved throughout the text. I suggest a careful revision by expert.
-Few suggestions I have mentioned in the main text pdf file. Please check
Title: suggestion: Genome-wide identification of GR-RBP genes and their expression analysis upon Aspergillus flavus infection in groundnut (Arachis hypogaea)
Abstract: -Good organization with results order.
-Indicate what is Ah.GR-RBP gene
-All genes should be same format and italic (Ah.GR-RBP.1, and AhGR-RBP.12-- not same style)
Keywords: use one more
1. Introduction: -The aim of the study should be underlined precisely and simultaneously and highlight why genome-wide identification analysis is important for this gene in groundnut. Rationale to be elucidated for the purpose of the study.
-Write something about Insilco study and their application in your rrsearch
2. Materials and Methods
-Well described
-Gene expression analysis needs more description. How many genes were selected for analysis and how they were selected?
-Which stage (age) cotyledon was used for treatment?
-Is there any strain for A. flavus spore?
-How the A. flavus was developed and maintained? Need description in the method
-Did you check any morphological changes of the samples after inoculation?
- L160: The inoculated cotyledons were immediately frozen in liquid nitrogen and stored at −80 °C until used for gene expression studies.------ immediately, not the correct timing.
Results
-Fig. 2. ---need high resolution and good readability
-Better to indicate centomeric position in the chromosomes
-What was the basis for the selection of nine genes for expression analysis?
-Better to show expression of reference gene ADH besides other genes
-How many biological and experimental replication were used in this experiment?
Discussion
-More relative discussion needed based on results obtained, especially for gene expression.
-Need recent references for microorganisms treatment to the cotyledon of groundnut and other plants.
My Recommendation about this article:
Revision needed.

Comments on the Quality of English LanguageModerate editing of English language required
Author Response
REVIEWER 2:
Dear Author,
I have an honor to review the manuscript entitled “Genome-wide identification and expression analysis of GR-RBPs during Aspergillus flavus infection in groundnut (Arachis hypogaea)” a research article submitted to the MDPI Journal, Agronomy. The authors of this manuscript documented GR-RBP gene families in groundnut using in silico-based genome-wide scanning analysis. They have identified chromosomal location, sub-cellular localization, and types of regulatory elements present in the putative promoter region. Further performed expression analysis of some of the selected genes after A. flavus infection. Overall, the experiments are performed well and the results are convincing. Thus, the presented results take up an important topic consistent with the profile of the Journal.
-However, even, manuscript is well organized and well described of the conception; I have some suggestions, which might improve the manuscript to make important to the wider audience.
Response: Thank you for your comments. We have addressed all the comments based on your suggestion.
- English should be improved throughout the text. I suggest a careful revision by an expert.
Response: We have improved the manuscript based on reviewer suggestion.
- Few suggestions I have mentioned in the main text pdf file. Please check
Response: We have improved the manuscript based on the reviewer's suggestion.
- Title: suggestion: Genome-wide identification of GR-RBPgenes and their expression analysis upon Aspergillus flavusinfection in groundnut (Arachis hypogaea)
Response: Thank you for your suggestion on the title. We have changed the title of the manuscript as per suggestion.
- Abstract: Good organization with results order.
Response: Thank you for your positive feedback.
- Indicate what is Ah.GR-RBP gene
Response: Ah.GR-RBP is the Arachis hypogaea GR-RBP gene. This has been made clear in the abstract in the revised manuscript.
- All genes should be in the same format and italic (Ah.GR-RBP.1, and AhGR-RBP.12-- not same style)
Response: Based on the reviewer's suggestion, we have used Ah.GR-RBP throughout the manuscript.
- Keywords:use one more
Response: We added secondary metabolite as one of the keywords in the updated manuscript.
- Introduction: The aim of the study should be underlined precisely and simultaneously and highlight why genome-wide identification analysis is important for this gene in groundnut. Rationale to be elucidated for the purpose of the study.
Response: Aflatoxin contamination caused by Aspergillus flavus in groundnut is a severe constraint to food safety and human health. Despite decades of research on groundnut, achieving success in developing aflatoxin-resistant varieties has been limited, partly due to the absence of highly resistant donors in the cultivated species. Targeting the transcriptional inactivation of susceptibility-associated genes holds the potential for developing durable resistance through focused breeding interventions. GR-RBPs have been reported as one of the candidate susceptibility-associated genes in maize upon A. flavus infection. However, their comprehensive genome-wide studies still need to be improved in groundnut. This is the first study to identify the GR-RBPs in groundnut and check their expression analysis upon A. flavus infection. This has been made clear in the introduction section.
- Write something about the Insilco study and its application in your research
Response: We have added the in-silico study and its application in the manuscript based on reviewer comments in lines 80-86.
- Materials and Methods
-Well described
-Gene expression analysis needs more description. How many genes were selected for analysis and how they were selected?
Response: We have updated the manuscript as suggested, including the information on selected genes in section 2.7.2, lines 188 to 189; we further explained the reason in results section 3.7 from lines 325 to 329.
- Which stage (age) cotyledon was used for treatment?
Response: Stored mature seeds were used for the treatment; we have described and mentioned the same in line 161 under section 2.7.1 in the updated manuscript.
- Is there any strain for A. flavusspore?
Response: A. flavus spores of toxigenic strain AF 11-4 were used for infection. We have described the same in line 164 to 166
- How the A. flavus was developed and maintained? Need a description in the method
Response: The description for maintenance was mentioned in detail from lines 165 to 173 in the updated manuscript.
- Did you check any morphological changes in the samples after inoculation?
Response: Differential fungal colonization was visually observed on the contrasting genotypes, as reported in previous studies. However, it was not studied in detail as the samples were dedicated to gene expression studies.
- L160: The inoculated cotyledons were immediately frozen in liquid nitrogen and stored at −80 °C until used for gene expression studies.------immediately, not the correct timing.
Response: We have rephrased the sentence in line 179 of the updated manuscript to make it more self-explanatory.
- Results
-Fig. 2. ---need high resolution and good readability
-Better to indicate the centomeric position in the chromosomes
Response: We have modified Figure 2 based on the reviewer's suggestion.
- What was the basis for selecting nine genes for expression analysis?
Response: The selected nine genes belong to GR-RBP2, which is reported to be upregulated in maize following the Aspergillus flavus infection. Since the present study deals with aflatoxin contamination in groundnuts caused by A. flavus, we have targeted only those genes that belong to GR-RBP2 in groundnuts for gene expression analysis. We have explained the same in the updated manuscript in results section 3.7 from lines 325 to 329 and in the discussion section 4.3 from lines 446 to 450.
- It is better to show expression of reference gene ADH besides other genes
Response: The expression data given in Figure 7 are relative expression in comparison to the reference gene, ADH3; the fold change values of each gene's relative expression were also mentioned in the trailing text in section 3.7
- How many biological and experimental replications were used in this experiment?
Response: In each experiment, we used three biological replicates with two technical replications. We have mentioned the same in the updated manuscript from lines 195 to 196.
- Discussion
-More relative discussion is needed based on results obtained, especially for gene expression.
Response: We have updated the manuscript as suggested in discussion section 4.3.
- Need recent references for microorganism treatment to the cotyledon of groundnut and other plants.
Response: We have updated the materials and methods section line 302 with recent references.
- My Recommendation about this article:
Revision needed.
Response: We have revised the manuscript based on the reviewer's comments and suggestions.

Reviewer 3 Report
Comments and Suggestions for Authors
In this study twenty-three GR-RBP genes were identified, and chromosomal location, sub-cellular localization, and regulatory elements in the putative promoter region were analyzed. Expression analysis revealed that Ah.GR-RBP.1, AhGR-RBP.12, Ah.GR-RBP.3 and Ah.GR-RBP.15 showed higher expression in the susceptible genotype. This paper would help to provide knowledge on potential candidate target genes for precise breeding interventions for aflatoxin mitigation in groundnut.
1. The abstract section should be further summarized.
2. I think these figures in this paper are not clear, improved version of figures should be re-drawed.
3. Please complain that why several GR-RPB genes did not contain a conserved glycine-rich domain and an RNA-recognition motif (RRM) in figure 4. Do they not belong to GR-RPB genes?
4. Cultivated peanut contains two set sub-genome A and B, I suggest that in figure 2 chromosomal location of GR-RBP should be presented in two lists.
5. In figure 5, could add more cis-elements involved in biotic stresses?
Comments on the Quality of English LanguageMinor editing of English language required
Author Response
REVIEWER 3:
This study identified twenty-three GR-RBP genes, and chromosomal location, sub-cellular localization, and regulatory elements in the putative promoter region were analyzed. Expression analysis revealed that Ah.GR-RBP.1, AhGR-RBP.12, Ah.GR-RBP.3 and Ah.GR-RBP.15 showed higher expression in the susceptible genotype. This paper would help to provide knowledge on potential candidate target genes for precise breeding interventions for aflatoxin mitigation in groundnut.
- The abstract section should be further summarized.
Response: We have rephrased and summarised the abstract as suggested by the reviewer.
- I think the figures in this paper are not clear; improved version of the figures should be re-drawed.
Response: We have redrawn all the Figures in the manuscript with high resolution.
- Please complain that why several GR-RPB genes did not contain a conserved glycine-rich domain and an RNA-recognition motif (RRM) in figure 4. Do they not belong to GR-RPB genes?
Response: We have selected 23 genes as the GR-RBP gene family based on comprehensive in-silico analysis. The multiple sequence alignment results revealed that GR-RBPs contain glycine-rich regions and two highly conserved RRM motif sequences (RNP1, eight amino acid residues, and RNP2, six amino acid residues). This has been presented in Figure S2. Although conserved domains exist, some genes lack conservation in the RRM motif and glycine-rich residues. This analogous trend was also noticed in previous studies on GR-RBPs identified in maize and grapes despite their classification within the GR-RBP family of proteins. We have updated the article in the discussion section, lines 665 to 669.
- Cultivated peanut contains two set sub-genome A and B, I suggest that in figure 2 chromosomal location of GR-RBP should be presented in two lists.
Response: We have updated the chromosomal location in Figure 2 with two sets of sub-genomes, A and B.
- In Figure 5, could we add more cis-elements involved in biotic stresses?
Response: Based on the reviewer's suggestion, we have updated Figure 5 with more cis-elements.
